# Retrospective study of toxoplasmosis prevalence in pregnant women in Benin and its relation with malaria

Magalie Dambrun[1]*, Célia Dechavanne[1], Nicolas Guigue[2], Valérie Briand[1¤], Tristan Candau[3], Nadine Fievet[1], Murielle Lohezic[1], Saraniya Manoharan[1], Nawal Sare[4], Firmine Viwami[4], François Simon[2], Sandrine Houzé[1,3], Florence Migot-Nabias[1]

1 Université de Paris, MERIT, IRD, Paris, France, 2 Service de Microbiologie, Assistance Publique—Hôpitaux de Paris, Groupe Hospitalier Saint-Louis-Lariboisière-Fernand-Widal, Paris, France, 3 Service de Parasitologie, Assistance Publique—Hôpitaux de Paris, Groupe Hospitalier Bichat Claude-Bernard, Paris, France, 4 Institut de Recherche Clinique du Bénin, Cotonou, Bénin

¤ Current address: Maladies Infectieuses dans les Pays à ressources Limitées (IDLIC team), INSERM U1219, Institut de Recherche pour le Développement, Université de Bordeaux, Bordeaux, France
* magalie.dambrun@u-paris.fr

**Data Availability Statement:** The data used were either extracted from the qualitative and quantitative datasets from the STOPPAM study on

## Abstract

### Background

Globally distributed with variable prevalence depending on geography, toxoplasmosis is a zoonosis caused by an obligate intracellular protozoan parasite, *Toxoplasma gondii*. This disease is usually benign but poses a risk for immunocompromised people and for newborns of mothers with a primary infection during pregnancy because of the risk of congenital toxoplasmosis (CT). CT can cause severe damage to fetuses-newborns. To our knowledge, no study has been conducted in sub-Saharan Africa on toxoplasmosis seroprevalence, seroconversion and CT in a large longitudinal cohort and furthermore, no observation has been made of potential relationships with malaria.

### Methods

We performed a retrospective toxoplasmosis serological study using available samples from a large cohort of 1,037 pregnant women who were enrolled in a malaria follow-up during the 2008–2010 period in a rural area in Benin. We also used some existing data to investigate potential relationships between the maternal toxoplasmosis serological status and recorded malaria infections.

### Results

Toxoplasmosis seroprevalence, seroconversion and CT rates were 52.6%, 3.4% and 0.2%, respectively, reflecting the population situation of toxoplasmosis, without targeted medical intervention. The education level influences the toxoplasmosis serological status of women, with women with little or no formal education have greater immunity than others. Surprisingly, toxoplasmosis seropositive pregnant women tended to present lower malaria infection during pregnancy (number) or at delivery (presence) and to have lower IgG levels to

malaria or generated by this study on toxoplasmosis. The newly acquired data on toxoplasmosis, linked to information on inclusion dates in the STOPPAM study, are available from the Open Science Framework database (https://osf.io/ut5ey/), DOI 10.17605/OSF.IO/UT5EY. Regarding the malaria data from the STOPPAM project, they are not currently publicly available. However, collaborations are encouraged and any researcher interested in exploring data may contact directly the researcher responsible for the local STOPPAM committee: Nicaise Tuikue Ndam (STOPPAM data, email: nicaise.ndam@ird.fr).

**Funding:** This study was supported by Institut de Médecine et d'Epidémiologie Appliquée (IMEA) (https://www.imea.fr) in the form of a grant to FMN [0602DIRmba].

**Competing interests:** The authors have declared that no competing interests exist.

*Plasmodium falciparum* Apical Membrane Antigen 1, compared to toxoplasmosis seronegative women.

## Conclusions

The high toxoplasmosis seroprevalence indicates that prevention against this parasite remains important to deploy and must be accessible and understandable to and for all individuals (educated and non-educated). A potential protective role against malaria conferred by a preexisting toxoplasmosis infection needs to be explored more precisely to examine the environmental, parasitic and/or immune aspects.

## Introduction

Toxoplasmosis is one of the most common parasitic diseases caused by *Toxoplasma gondii*, an intracellular protozoan parasite belonging to the phylum *Apicomplexa*. This parasite can infect all warm-blooded animals. Approximately 30% of the human population is infected worldwide, but the prevalence of infection is variable between both areas and communities, according to climate, lifestyle and diet [1]. Toxoplasmosis is the third burden of foodborne illness in Europe for WHO [2], while the American Centers for Disease Control and Prevention (CDC) considers it a neglected disease due to the scant attention paid to it in terms of surveillance, prevention and/or treatment [3, 4].

Toxoplasmosis is usually benign. However, it has adverse consequences in immunocompromised people and fetuses-newborns from women who have contracted toxoplasmosis during their pregnancy. Congenital toxoplasmosis (CT) is the result of mother-to-child vertical transmission of the parasite, and the risk increases with gestational age. The severity of the encountered disorders is inversely related to the pregnancy period at maternal infection. The disorders may range from severe abnormalities (mainly neurological and ophthalmic) or abortion during the first trimester of gestation, to manifestations of variable severity during the second trimester and asymptomatic traces at the third trimester [1, 5, 6]. Other risk factors, including immune and genetic host factors as well as characteristics of the *T. gondii* strain, are involved in the severity of the disease [6]. In *T. gondii* non-immune pregnant women, treatment is recommended as soon as the toxoplasmosis contamination is proven in order to reduce the risk of parasite transmission to the fetus and of sequelae in neonates. The contamination can be detected indirectly by the seroconversion of the woman, which is defined by the appearance of specific IgG directed to *T. gondii*. Treatments are given depending on the stage of pregnancy. For example, in France, spiramycine 9 million units/day is recommended during the first trimester [6, 7] whereas pyrimethamine 50mg/day associated with sulfadiazine 3x1g/day and folinic acid 50mg/week are recommended from the second trimester to the end of pregnancy [6].

The most effective way to detect seroconversion during pregnancy, and then to apply an antiparasitic treatment to prevent CT, is to implement a national serological screening program of pregnant women. Although this practice would be effective and less financially burdensome than monitoring pregnancy [8–10], it is rarely instituted globally [5, 6, 11] and therefore hygiene and diet guidelines remain the most used means of first-line prevention [6, 12]. The lack of systematic serological follow-up of pregnant women limits the informative data on actual seroprevalence and the incidence of seroconversion, especially as the infection is asymptomatic, potentially hiding a considerable number of CT cases. Although many

countries lack adequate data, some epidemiological meta-analyses have been performed, leading to global estimates of 33.8% for acquired toxoplasmosis among pregnant women [13], 0.6%–1.1% for the seroconversion rate [14] and 190,100 cases/year (1.5 cases per 1,000 births) for the overall CT incidence [15].

Although data on human toxoplasmosis are less numerous in Africa than in some other countries [13, 16], they show that the seroprevalence rate, recently estimated at a global value of 48.7%, undergoes a great variability between countries [13]. Indeed, the seroprevalence rates recorded during the last decade in West Africa ranged from 20% to 70%, depending on countries [17–20], despite different methodologies used. A downward trend is recorded compared to previous periods, thus the toxoplasmosis seroprevalence decreased from 74% in 2006 [21] to 40% in 2016 [19] in Ghana, and from 54% in 1993 [22] to 49% in 2012 [23] in Cotonou, the economic capital of Benin. A recent meta-analysis in Benin indicated a toxoplasmosis mean seroprevalence of 47% among pregnant women over the 1990–2018 period [24]. Inter-regional variability is also observed in Benin with seroprevalence varying from 30% in 2011 in the Atacora department in the north [25] to 49% in 2012 in the Littoral department in the south including Cotonou [23], and 36% in 2016 in the Atlantic department in the west [26].

As with many other sub-Saharan countries, Benin is also impacted by malaria. This disease is caused by *Plasmodium spp.*, another *Apicomplexa* parasite. The main species involved is *Plasmodium falciparum*, known to impair the fetal development in case of pregnancy-associated malaria. The latest WHO report indicated that 11 million pregnant women living in sub-Saharan malaria-endemic areas were exposed in 2018 to malaria infection. This resulted in 16% of the cases of low-birth-weight children recorded in these areas [27]. WHO recommends an intermittent preventive treatment against malaria during pregnancy (IPTp) using a sulfadoxine-pyrimethamine (SP) combination (IPTp-SP), the same molecules as for toxoplasmosis treatment but dosed differently. Since October 2012, IPTp-SP has been recommended as soon as possible from the second trimester of pregnancy, with at least three doses spaced one month apart until delivery [27].

To date, there is no description of toxoplasmosis seroprevalence, seroconversion and CT rates on large cohorts of pregnant women and their infants in West Africa. To our knowledge, only one study performed in Ghana addressed the diagnosis of *Plasmodium* and *Toxoplasma* co-infections at delivery among fewer than 100 pregnant women recruited during the third trimester of pregnancy [28].

In this study, we provide additional information on toxoplasmosis during pregnancy in Benin by performing a retrospective serological study using plasmas of pregnant women from a previous study on gestational malaria. This offered a unique opportunity to explore the potential relationships between maternal toxoplasmosis serological status and *Plasmodium* malaria and IPT-SP in pregnancy.

## Material and methods

### Population group

The women under study participated in the "Strategy To Prevent Pregnancy-Associated Malaria" (STOPPAM) project, implemented from 2008 to 2010 in southern Benin, where the climate is subtropical [29, 30]. The STOPPAM project was designed to explore the relationships between the timing of malaria infection during pregnancy and clinical and immunological consequences for both mothers and infants.

Women were enrolled in three dispensaries from one semi-rural and two rural sites, in the district of Comé in the Mono province, located 70km west of Cotonou. The main occupations of the population are farming, fishing and trading. The inclusion criteria were: living for more

than 6 months within 15 km from the dispensary, having a gestational age under 24 weeks and having planned to deliver at the hospital. A total of 1,037 women were enrolled, and 982 were followed-up through pregnancy, including 891 until delivery [29].

At each visit (e.g., monthly antenatal care (ANC) visit, in case of an emergency and at delivery), clinical and obstetrical data as well as biological samples (including blood samples which were used for this study) were collected.

The STOPPAM project followed WHO recommendations in effect at that time, so women received IPTp, iron and folic acid as per national guidelines. When diagnosed malaria-infected, women were treated with quinine or SP following the recommended WHO process [29].

### Design of the serological study on toxoplasmosis

**Determination of the toxoplasmosis seroprevalence rate.** Serology against *T. gondii* was carried out on the available ethylenediaminetetraacetic acid (EDTA) frozen plasma samples obtained at each woman's enrollment into the STOPPAM project (Fig 1), either at the first ANC visit or at the consultation following it, usually after one month. ToRC (Toxoplasmosis, Rubella, Cytomegalovirus) IgG on the BioPlex® 2200 System (Bio-Rad) was assessed according to the manufacturer's recommendations. It is a multiplex immunoassay for quantitative detection of anti-*T. gondii* and anti-rubella IgG and for qualitative detection of anti-CMV (cytomegalovirus) IgG in a single reaction from a single serum or plasma sample. Anti-rubella and anti-CMV IgG results are not presented here. A specific software associated with the Bio-Plex 2200 automated analyzer (Bio-Rad) was used, and *T. gondii* results were expressed according to the following thresholds: results $\leq$ 9 IU/mL were negative, those between 10–11 IU/mL were doubtful, and results $\geq$ 12 IU/mL were positive. In case of positive results at the inclusion, the woman was considered as immunized and no further IgG assay was performed.

**Determination of the toxoplasmosis seroconversion rate during pregnancy.** For toxoplasmosis seronegative women at inclusion, a measurement of anti-*T. gondii* IgG was performed on available samples corresponding to the end of the study, usually at delivery, or the previous consultation not exceeding one month before delivery (Fig 1). Platelia™ TOXO IgG (Bio-Rad), an indirect ELISA immunoassay for quantitative determination of anti-*T. gondii* IgG in human serum or plasma, was used according to the manufacturer's recommendations. The absorbance was read at 450/620nm on Asys UVM 340 spectrophotometer (Biochrom). Results corresponding to IgG titers <6 IU/mL were negative; they were equivocal for IgG titers comprised between 6 and 9 (excluded) IU/mL and positive for IgG titers $\geq$ 9 IU/mL.

When the test was positive at the end of pregnancy for a woman previously diagnosed as seronegative, Platelia™ TOXO IgG was carried simultaneously on both inclusion and exit study plasma samples of each individual to check the transition from negativity to positivity using the same immunoassay. For the rare women with discrepant results on the sample of the end of the study (i.e., the plasma of the end of the study was tested positive for the first assay and then negative during the second run, both runs being assessed with Platelia™ TOXO IgG), LDBIO Toxo II IgG confirmation (LDBIO Diagnostics) was used according to the manufacturer's recommendations. This immunoblot qualitative assay is proposed to confirm positive or equivocal results obtained by classical serological tests. Presence of specific anti-*T. gondii* IgG was confirmed if at least three bands were observable at either 30, 31, 33, 40 or 45 kDa, including the band at 30 kDa in all cases.

Pregnant women who evolved from negative to positive anti-*T. gondii* IgG between inclusion and exit of the study were analyzed more specifically. Namely, all available plasma samples that had been sequentially collected at each visit (an average of six visits) during pregnancy

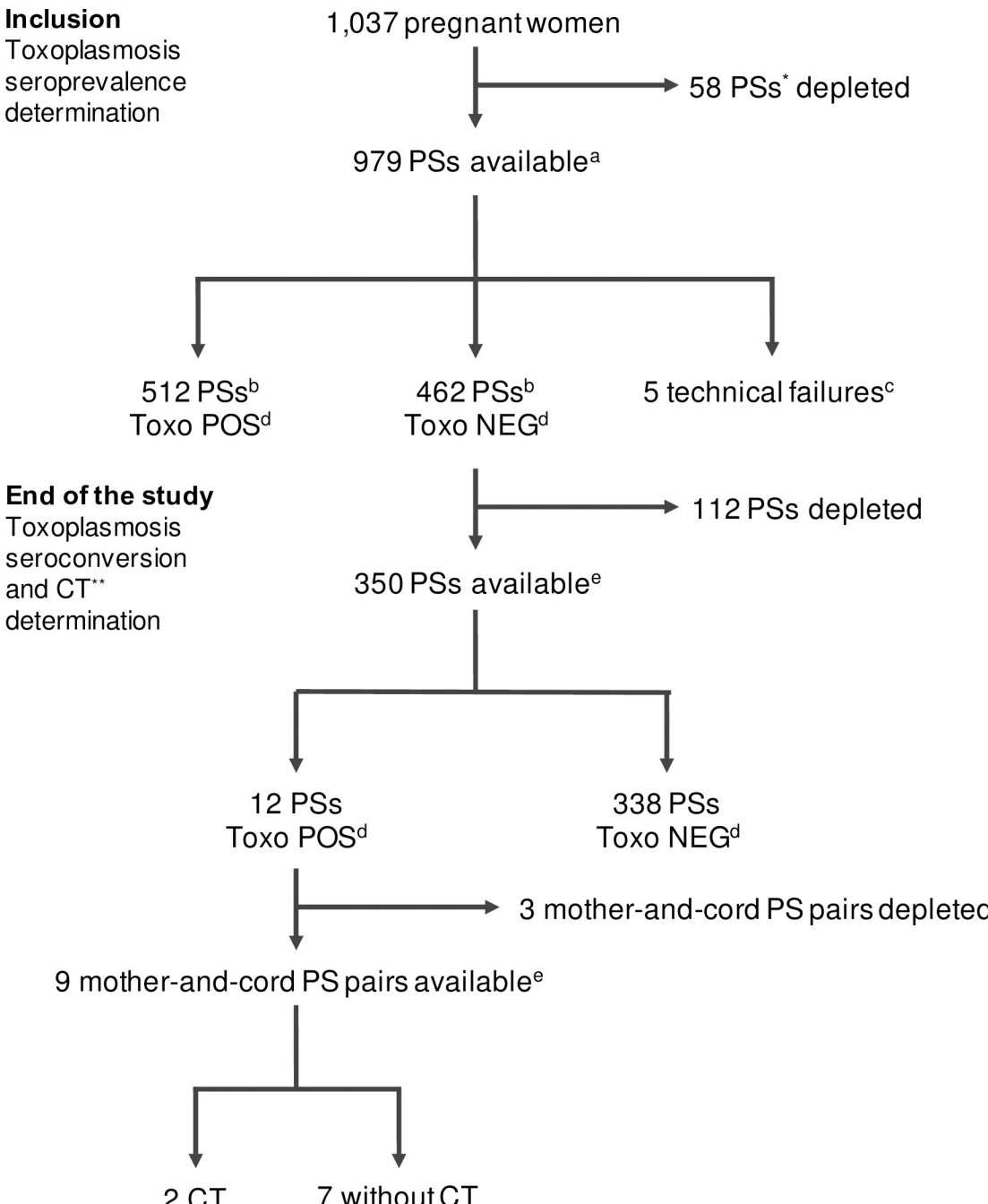

**Fig 1. Flow chart of the serological study on toxoplasmosis.** [*]: PS = plasma sample. [a]: available plasma samples at inclusion, or during the first antenatal care visit or the next visit, not exceeding one month after the first one. [b]: diluted and undiluted plasma samples (see Materials and Methods: "Methodological constraints"). [c]: poor quality plasma samples with too much cell debris. [d]: serological toxoplasmosis results positive (Toxo POS) or negative (Toxo NEG). [e]: available plasma samples corresponding to the end of the study, usually at delivery or during a previous visit not exceeding one month before delivery. [**]: CT = congenital toxoplasmosis.

were tested in a single experiment with Platelia™ TOXO IgG, together with the initial analyzed samples (inclusion and exit) to evaluate the seroconversion period. The determination was refined by detecting anti-*T. gondii* IgM on the plasma samples corresponding to the period

that circumscribed the IgG seroconversion. For this purpose, the qualitative Platelia™ TOXO IgM (Bio-Rad) detection assay was used, according to the manufacturer's recommendations. The absorbance was read at 450/620nm under the same conditions as Platelia™ TOXO IgG. Results were expressed as a ratio, which may be negative ($<$0.8), equivocal (from 0.8 to $<$1) or positive ($\geq$ 1).

The IgG avidity test described by Robert-Gangneux *et al.* [31] was applied using Platelia™ TOXO IgG (Bio-Rad) for one woman (pregnant woman #1, or PW1) with equivocal IgG results throughout the pregnancy. This method allows for the determination of an avidity index expressed in percentage depending on the age of infection: a high avidity index indicates a formerly acquired immunity ($\geq$ 10 months) and a lower avidity index indicates a possible recently contracted toxoplasmosis infection. Then it is even possible to estimate the period of infection to be less than 3 months, 3 to 5 months and 5 to 10 months.

**Detection of cases of Congenital Toxoplasmosis (CT).** The detection of CT in newborns was made for the group of women who had seroconverted. Qualitative Platelia™ TOXO IgM (Bio-Rad) detection assay was used on cord blood plasmas, according to the manufacturer's recommendations. Also, an immunoblot toxoplasmosis assay (Toxoplasma Western Blot IgG IgM, LDBIO Diagnostics) was performed for the comparison of mother-and-cord immunological IgG profiles. Maternal plasma samples (from circulating blood at delivery) and fetal plasma samples (from cord blood) were used. According to the manufacturer's recommendations, maternal and cord plasma samples were incubated on two separate and contiguous strips from the same nitrocellulose transfer membrane on which *T. gondii* antigens separated by electrophoresis were bound. A polyclonal goat anti-human IgG coupled with alkaline phosphatase was used as conjugate. CT is suggested by the presence of a band in the cord but not the maternal plasma at a molecular weight less than 120 kDa. Because of insufficient plasma quantities, a similar immunoblot toxoplasmosis assay using an IgM conjugate was not performed.

## Methodological constraints

This sero-epidemiological study was constrained by sample availability, quantity and quality. First, some plasma samples were fully utilized following the initial malaria research program (Fig 1). Then, to avoid damaging the analysis automat for the ToRC assay, due to insufficient volume or too high viscosity, a number of enrollment plasma samples were diluted 1/3 in washing buffer, as recommended by the manufacturer. More precisely, this adaptation concerned 348 out of the 979 processed samples (35.5%). Five of them ended in technical failure. Without applying any dilution corrective factor, these samples yielded 189 seropositive and 154 seronegative results. The interpretation of the seronegative results could have been tricky, due to the 0–9 scale of values corresponding to the negativity threshold of the test ($\leq$ 9 IU/mL), but matching with the serological results at study exit (realized with Platelia™ TOXO IgG on non-diluted samples) allowed for confirmation of seronegativity for 105/154 of them. The remaining 49 plasma samples at exit were not available because having been depleted: it was decided to consider these 49 plasma samples as seronegative, insofar as their inclusion in or exclusion from the analysis had little influence on the seronegativity rate of the entire cohort (n = 462/974, 47.4% *vs.* n = 413/925, 44.6%).

## Statistical analysis

Firstly, we calculated the prevalence of maternal toxoplasmosis based on IgG levels to *T. gondii* at the start of the study; IgG titers (UI/mL) were transformed into binary variables (seronegative/seropositive) according to the thresholds described above. Toxoplasmosis seroconversion was defined as the transition from seronegativity at inclusion to seropositivity at the end of the

follow-up. The timing of seroconversion, as well as maternal and newborn characteristics in case of seroconversion were described. Finally, the prevalence of CT (95% CI), defined as mother-and-cord immunological IgG different profiles, was calculated.

Maternal and newborn characteristics associated with *T. gondii* serological maternal status at inclusion were determined using chi-square test or t-tests. For these analyses, women who had seroconverted during pregnancy (n = 12) were excluded from these analyses and were considered as a separate group.

The following characteristics were tested for their possible association with serological toxoplasmosis maternal status: 1) socio-epidemiological data: village of residence, season at enrolment, mother's education level, bednet possession and number of medical visits (ANC and emergency) during follow-up; 2) maternal clinical data: age, gravidity, gestational age at enrolment and at delivery determined by ultrasound [32]; and 3) neonatal data: prematurity (<37 weeks gestation), stillbirth, birth weight and low birth weight (<2,500 grams).

Secondly, we assessed the association between *T. gondii* serological maternal status at inclusion and maternal malaria infection (at inclusion, during pregnancy and at delivery). Malaria infection during pregnancy was defined as at least one positive thick blood smear (TBS) during ANC and emergency visits. Malaria infection at delivery was considered positive when parasites were detected either in peripheral or placental blood. A logistic univariate analysis was performed to assess this association, and a following multivariate logistic model adjusted on maternal socio-demographic and clinical characteristics was conducted to ensure an association existed. At least, the mean of maternal plasma antibody levels to the Apical Membrane Antigen 1 of *P. falciparum* (anti-*Pf*AMA1 IgG), both at inclusion and delivery, was log-transformed and compared according to the *T. gondii* serological maternal status at inclusion. IgG directed to *Pf*AMA1, which is a merozoite antigen from the erythrocytic asexual stages of *P. falciparum* and also a conserved apicomplexan protein, were measured by Enzyme-Linked ImmunoSorbent Assay (ELISA) as previously described [33]. Associations were then assessed using a linear univariate analysis followed by a multivariate linear regression model to account for potential confounding factors such as maternal socio-demographic and clinical characteristics. The differences of anti-*Pf*AMA1 IgG levels between inclusion and delivery were established using a multivariate mixed model where antibody levels at inclusion and at delivery were paired for each woman.

Data were analyzed with Stata$^®$ Software, Version 13 (StatCorp LP, College Station, TX, USA) and the graphics were done using Graph Pad Prism (Version 8.1.2).

### Ethics

The STOPPAM project was approved by the ethics committees of the Research Institute for Development (IRD) in France and of the Science and Health Faculty (University of Abomey Calavi) in Benin. Informed written consent with the possibility of withdrawing from the study at any time was obtained from all women before enrollment. The retrospective sero-epidemiological investigation of toxoplasmosis during pregnancy performed on the STOPPAM cohort was approved by the STOPPAM project committee. All the methods were performed in accordance with the institutional guidelines and regulations pertaining to research involving humans.

## Results

### Toxoplasmosis seroprevalence, seroconversion and CT

The different steps of the serological survey are presented in Fig 1. Among plasma samples collected at the inclusion of 1,037 pregnant women in the STOPPAM study, 979 samples from the first visit were available for the determination of toxoplasmosis seroprevalence. Excluding

5 samples due to technical failure, 512 and 462 women were found serologically positive and negative to toxoplasmosis, respectively. Therefore, the toxoplasmosis IgG seroprevalence was 52.6% (512/974). Among the 462 toxoplasmosis seronegative women at enrolment, 350 samples at delivery (or the preceding period when plasma samples at delivery were depleted) corresponding to 75.7% of them, were available to establish the toxoplasmosis seroconversion rate. Twelve women were found to have contracted a primo-infection by *T. gondii* between these two measurement points, their serological IgG toxoplasmosis status having changed from negative to positive. Therefore, a toxoplasmosis seroconversion rate of 3.4% (12/350, 95% CI [0.0178; 0.059]) was observed when considering seronegative pregnant women with available plasma samples at delivery. Among the 12 cases identified with seroconversion during pregnancy, 9 mother-and-cord plasma pairs were available for highlighting potential cases of CT. Since 2 CT cases could be observed only based on IgG immunoblot results (qualitative Platelia™ TOXO IgM cord blood results being negative and IgM immunoblotting could not be performed), the congenital toxoplasmosis rate was 0.2%, corresponding to 2/835 live births (95% CI [0.0003; 0.086]), i.e., a ratio of 2.39 per 1,000 live births, among the initial group of 974 women with known toxoplasmosis serological status; knowing that 19 births were twins, 163 deliveries remained unrecorded, for 816 women (i.e., 835–19) among the whole study group of 979 women. These 163 situations were distributed into 80 women lost to follow-up or having stopped the follow-up (49.1%), 35 stillbirths (21.5%), 32 miscarriages (19.6%) and 16 women whose pregnancy has been invalidated (9.8%).

## Women's and newborns' characteristics according to their toxoplasmosis serological profile

Environmental and clinical characteristics recorded during the STOPPAM study for the mothers and their newborns were used for the present analysis. It appears that *T. gondii* seropositive women were older (27.6 ± 6.2 years *vs*. 25.1 ± 6.0 years; *P*<0.001) and had higher gravidity (3.7 ± 2.2 *vs*. 3.1 ± 1.8; *P* = 0.003) than those who remained seronegative all along their pregnancy (Table 1). Gestational age at delivery did not differ between the two groups. Regarding living conditions, no difference was observed between groups of women regarding either their rural or semi-rural location or their bednet possession. Also, no difference regarding presence of anemia was evidenced between groups. Finally, *T. gondii* seropositive women had completed a lower education level than *T. gondii* seronegative ones (*P* = 0.006).

On their own, newborns did not differ in terms of prematurity, stillbirth, mean birth weight or low birth weight prevalence according to the toxoplasmosis serological status of their mothers during pregnancy (Table 2).

## Seroconversion maternal group and infants

Table 1 also presents the clinical data of women who contracted toxoplasmosis during pregnancy (n = 12). Their mean age at inclusion was 28.6 ± 5.5 years, their mean number of previous pregnancies was 3.8 ± 1.9; they were enrolled at 15.9 ± 5.9 gestation weeks and gave birth at 39.9 ± 1.6 gestation weeks. For 7 out of these 12 women, plasma samples at inclusion were diluted 1/3 for the ToRC assay, and their negativity confirmed later using the Platelia™ TOXO IgG assay on non-diluted plasma samples. Among infants born to this seroconversion maternal group, one case of prematurity and two cases of low birth weight were recorded (Table 2).

Ultrasonography performed during the follow-up did not reveal any abnormality in fetuses suspected of CT.

Since an average of six blood samples were collected for this group during pregnancy, this allowed over time anti-*T. gondii* IgG measurements for an individual estimate of the

**Table 1. Clinical and environmental characteristics of pregnant women according to their *T. gondii* serological status.**

| Maternal characteristics | *T. gondii* seropositive group (n = 512) | | *T. gondii* seronegative group (n = 450) | | *P* [a] | *T. gondii* seroconverting group (n = 12) | |
|---|---|---|---|---|---|---|---|
| | | | Serological toxoplasmosis status at inclusion | | | | |
| | n | Mean (± SD) or % | n | Mean (± SD) or % | | n | Mean (± SD) or % |
| Age (years) | 505 [b] | 27.6 (± 6.2) | 443 [c] | 25.1 (± 6.0) | **<0.001*** | 12 | 28.6 (± 5.5) |
| Gravidity (n) | 512 | 3.7 (± 2.2) | 450 | 3.1 (± 1.8) | **0.003** | 12 | 3.8 (± 1.9) |
| Primigest | 77 | 15.0% | 101 | 22.4% | **<0.001*** | 1 | 8.3% |
| Secondigest | 110 | 21.5% | 107 | 23.8% | | 2 | 16.7% |
| Multigest | 325 | 63.5% | 242 | 53.8% | | 9 | 75.0% |
| Gestational age at enrolment (weeks)[d] | 512 | 16.1 (± 4.7) | 450 | 16.8 (± 4.9) | **0.03*** | 12 | 15.9 (± 5.9) |
| Gestational age at delivery (weeks) | 440 [b] | 39.2 (± 2.9) | 395 [c] | 39.4 (± 2.4) | 0.31* | 11 [e] | 39.9 (± 1.6) |
| Living site | | | | | | | |
| Rural sites (Akodeha and Ouedeme Pedah) | 289 | 56.5% | 261 | 58.0% | 0.63 | 8 | 66.7% |
| Semi-rural site (Comé) | 223 | 43.5% | 189 | 42.0% | | 4 | 33.3% |
| Maternal education | | | | | | | |
| None | 317 | 62.0% | 228 | 50.7% | **0.006** | 8 | 66.7% |
| Partial primary | 98 | 19.1% | 107 | 23.8% | | 4 | 33.3% |
| Complete primary | 34 | 6.6% | 42 | 9.3% | | - | - |
| Beyond primary | 63 | 12.3% | 73 | 16.2% | | - | - |
| Number of visits (ANC + urgency) | 512 | 5.31 (± 2.2) | 450 | 5.12 (± 2.2) | 0.19* | 12 | 5.2 (± 1.8) |
| Season at enrolment | | | | | | | |
| Dry season | 212 | 41.4% | 206 | 45.8% | 0.17 | 2 | 16.7% |
| Not dry season | 300 | 58.6% | 244 | 54.2% | | 10 | 83.3% |
| Bednet possession | | | | | | | |
| Yes | 166 | 32.4% | 139 | 31.0% | 0.61 | 4 | 33.3% |
| No | 346 | 67.6% | 311 | 69.0% | | 8 | 66.7% |
| IPTp-SP dose | | | | | | | |
| No dose | 36 | 7.0% | 26 | 5.8% | 0.61 | 0 | 0% |
| One dose | 38 | 7.4% | 39 | 8.7% | | 0 | 0% |
| Two doses | 435 | 85.0% | 384 | 85.3% | | 12 | 100% |
| Three doses | 3 | 0.6% | 1 | 0.2% | | 0 | 0% |
| Anemia | | | | | | | |
| At inclusion | 311/509[b] | 61.1% | 271/446[c] | 60.8% | 0.91 | 6 | 50.0% |
| At delivery | 144/328[b] | 43.9% | 138/295[c] | 46.8% | 0.47 | 3/7[e] | 42.8% |

[a]: *P* value of the t-test

* or chi-squared test

*P* < 0.05 in bold.

[b]: available values out of 512 women.

[c]: available values out of 450 women.

[d]: gestational age determined by ultrasound.

[e]: available values out of 12 women.

seroconversion period, as illustrated in Fig 2. Anti-*T. gondii* IgM levels were measured on samples circumscribing the seroconversion time-point, identifiable by the change of anti-*T. gondii* IgG results from negative to positive, in order to strengthen the diagnostic argument. Results were negative in all cases except one for which IgM levels changed from borderline to positive (PW5). Three women may have seroconverted towards the end of the first trimester of pregnancy (PW1, PW2 and PW3). For one of them (PW1), anti-*T. gondii* IgG doubtful results

**Table 2. Clinical characteristics of newborns according to the *T. gondii* serological status of their mothers during pregnancy.**

| Newborn clinical characteristics | *T. gondii* seropositive mothers (n = 440[a]/512) | | *T. gondii* seronegative mothers (n = 395[a]/450) | | *P* [b] | *T. gondii* seroconverting mothers (n = 11[a]/12)[c] | |
|---|---|---|---|---|---|---|---|
| | n | Mean (± SD) or % | n | Mean (± SD) or % | | n | Mean (± SD) or % |
| Prematurity (< 37 weeks) | 44 | 10.0% | 41 | 10.4% | 0.86 | 1 | 9.1% |
| Stillbirth | 21 | 4.8% | 14 [d] | 3.5% | 0.38 | 0 | 0% |
| Birth weight (g) | 435 [e] | 2953 (± 536) | 385 [f] | 2961 (± 503) | 0.88 | 11 | 3040 (± 631) |
| Low birth weight (< 2500g) [g] | 62 | 14.2% | 47 | 12.2% | 0.39 | 2 | 18.2% |

[a]: available newborn values.

[b]: *P* value of the univariate logistic regression.

[c]: the values mentioned here are indicative.

[d]: 1 missing value.

[e]: 5 missing values.

[f]: 10 missing values.

[g]: calculated on the number of infants with documented birth weight.

needed an assessment using an IgG avidity test. Low and crescent results from the first to the third measure led to a broad estimate for the seroconversion period covering the end of the first trimester and the whole second trimester (Fig 2). Two women were infected during the 6th month (PW4 and PW5), and their newborns had low birth weight, including one (PW5) who was born prematurely with no apparent link with toxoplasmosis. Seven women (from PW6 to PW12) seroconverted during the third trimester of pregnancy, including two (PW9 and PW10) during the end of the eighth month and two (PW11 and PW12) during the ninth month. The two congenital toxoplasmosis cases that were identified were the result of a

| Trimester | 1st trimester | | | | | | | | | 2nd trimester | | | | | | | | | | | | | 3rd trimester | | | | | | | | | | | | |
|---|---|---|---|---|---|---|---|---|---|---|---|---|---|---|---|---|---|---|---|---|---|---|---|---|---|---|---|---|---|---|---|---|---|---|---|
| Month of pregnancy | ... | 2 | | | 3 | | | | | 4 | | | 5 | | | | 6 | | | | 7 | | | 8 | | | | | 9 | | | | | | |
| Term (amenorrhea week) | 7 | 8 | 9 | 10 | 11 | 12 | 13 | 14 | 15 | 16 | 17 | 18 | 19 | 20 | 21 | 22 | 23 | 24 | 25 | 26 | 27 | 28 | 29 | 30 | 31 | 32 | 33 | 34 | 35 | 36 | 37 | 38 | 39 | 40 | 41 |
| PW1 | | | | | | | | | | | | | | | | | | D/N | | | D/N | | | | D/N | | | | | | | D/N | D/N | | |
| PW2[a] | | | | | | N/N | | | | | P/N | | | | | P | | | | | P | | | | P | | | | P | | | | | | P |
| PW3 | | | | | | | | N/N | | | P/N | | | | | | | | | | P | | | P | | | | | P | | | | | | P |
| PW4[b] | | | | N | | | | | | x | | | | N | | | x | | | | N/N | | | | | | x | | | | | x | P/N | | |
| PW5[b,c] | | | | | | | N | | | | | N | | | | | N | | | | N/D | | | | | | | | | P/P | | P | | | |
| PW6[d] | | | | | | | | | | | | | N | | | | | N | | | N/N | P/N* | | | | | | | | | | | | | |
| PW7 | | N | | | N | | | | | | | | | | | N | | | | | | | | | N/N | | | | | | | | P/N | | |
| PW8 | | | | | | | | | | | | N | | | | | N | | | | | N | | | | | N/N | | | | | P/N | | | P |
| PW9[a] | | | | | | | | | | N | | | | N | | | | N | N | | | | | | N | | | | N/N | | | | P/N | | |
| PW10 | | | | | | | | | | | | | | | | | N | | | | | | | | N | | | | N/N | | | | P/N | | |
| PW11 | | | | | | | | N | | | | | N | | | | | N | | | | | N | | | | | N | | | | N/N | | P/N | |
| PW12 | | | | N | | | | | N | | | N | | | | | | | | | | | | | | | | | | | | N/N | | P/N | |

**Fig 2. Schematic representation of the timing of toxoplasmosis seroconversion in the subgroup of 12 pregnant women.** The representation of the gestational period begins from the second month of pregnancy. Each pregnant woman (PW) corresponds to a line. For each line, the antenatal care visits (ANC) are represented either by one or two serological results or by an "x" when the sample was not available; emergency visits (visit excluding ANC) are represented by an asterisk (*). Each follow-up starts by an inclusion and finishes by delivery excepted for the PW6 woman who was lost to follow-up after her emergency visit. On each line, qualitative anti-*T. gondii* IgG and IgM results are mentioned as negative (N), positive (P) or doubtful (D); anti-*T. gondii* IgG is mentioned in all cases and is followed (/) by anti-*T. gondii* IgM result when it has been achieved, i.e., IgG or IgG/IgM. Colored areas correspond to confirmed (grey) or estimated (hatched grey) seroconversion period. Regarding possible CT cases, mother-and-cord immunological IgG profiles were identical except for PW mentioned in bold where distinct profiles were in favor of CT. Mother-and-cord immunological IgG profiles were not performed for underlined PWs because of unavailability of either mother or cord blood sample. [a]: maternal seroconversion associated with a strong CT suspicion. [b]: maternal seroconversion associated with newborn low birth weight. [c]: maternal seroconversion associated with premature birth. [d]: case of maternal seroconversion lost to follow-up before delivery.

maternal infection occurring between the third and fourth months of pregnancy for one case (PW2) and at the end of the pregnancy for another (PW9).

## Association between *T. gondii* serological status and P. falciparum infection during pregnancy

Fig 3 presents the percentages of malaria-infected women at three time points (inclusion, follow-up and delivery). Presence of a *Plasmodium* infection at enrollment did not differ between *T. gondii* seronegative ($n_{inclusion}$ = 84/450, 18.7%) and seropositive ($n_{inclusion}$ = 78/512, 15.2%, $P$ = 0.16) pregnant women. Nevertheless, during follow-up and delivery, *T. gondii* seronegative women had more malaria infections than *T. gondii* seropositive ones ($n_{follow-up}$ = 186/450, 41.3% *vs.* $n_{follow-up}$ = 179/512, 35.0%, $P$ = 0.04 and $n_{delivery}$ = 48/450, 10.7% *vs.* $n_{delivery}$ = 32/512, 6.3%, $P$ = 0.01). However, although this association did not reach significance in a multivariate analysis during the follow-up, this observation was supported by the negative association between the toxoplasma serological status and presence of a malaria infection at delivery (OR = 0.531 and $P$ = 0.012; Table 3). This result shows that women seropositive for *T. gondii* develop less malaria infections at delivery than women seronegative for *T. gondii.* This analysis has been conducted in a multivariate model meaning that the negative association between *T. gondii* serology and malaria infections at delivery is independent from the effect of adjustment variables on malaria infections at delivery. This was namely the case for IPTp which did not impact on the presence of a malaria infection at delivery, whatever the number of IPTp doses administered during pregnancy (S1 Table).

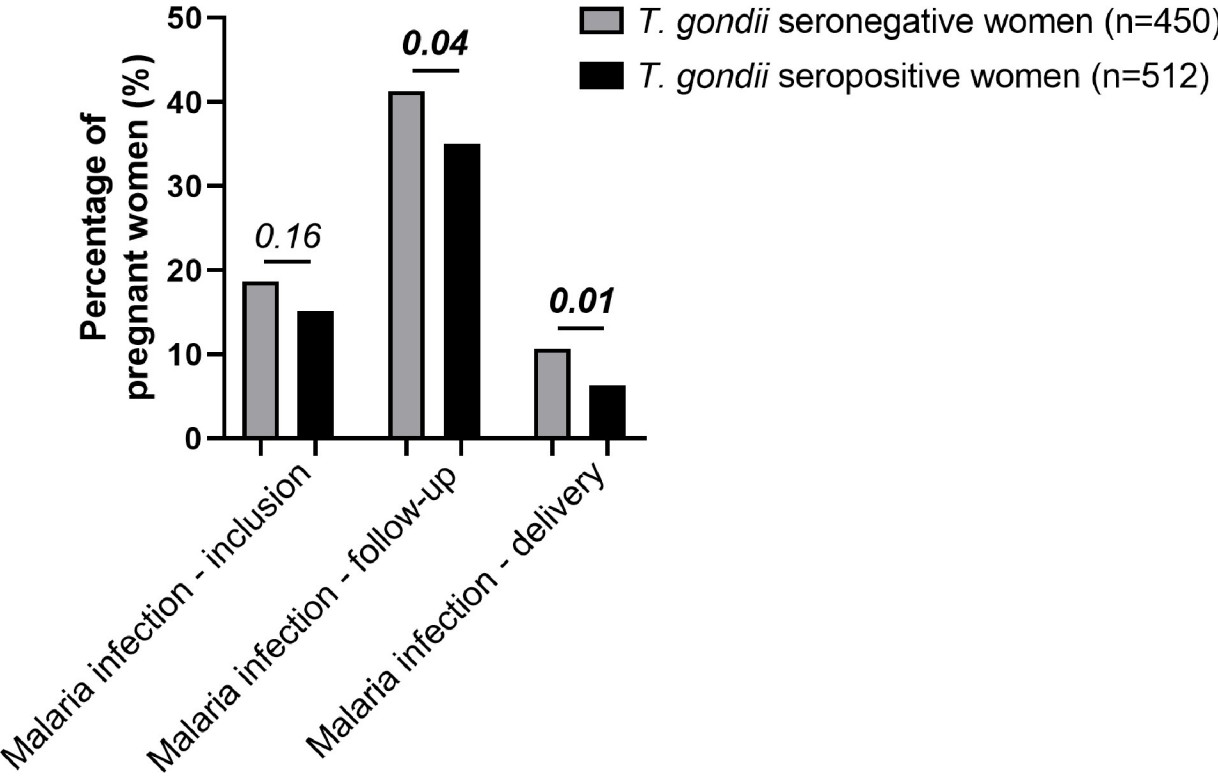

**Fig 3. Maternal malaria events in relation to the *T. gondii* serological status.** Maternal malaria infection determined by TBS during the STOPPAM study was used to compare pregnant women according to their toxoplasmosis serological status. *T. gondii* seronegative women are in the grey bar (n = 450) and *T. gondii* seropositive women in the black bar (n = 512).

**Table 3. Multivariate analysis exploring the association between malaria infection and *T. gondii* maternal serological status.**

| Independent variables | Categories | Malaria infection[a] during the follow-up (n = 948) | | | Malaria infection[a] at delivery (n = 948) | | |
|---|---|---|---|---|---|---|---|
| | | Odds Ratio[b] | [95% CI] | P value[c] | Odds Ratio[b] | [95% CI] | P value[c] |
| *T. gondii* positive serological status[d] | | 0.835 | [0.626; 1.116] | 0.223 | 0.531 | [0.324; 0.87] | **0.012** |
| Adjustment variables: | | | | | | | |
| Maternal age[e] | [15–22] | | | | | | |
| | [23–29] | 0.717 | [0.488; 1.053] | 0.090 | 0.768 | [0.387; 1.526] | 0.451 |
| | [30–35] | 0.434 | [0.285; 0.661] | **<0.001** | 1.198 | [0.592; 2.421] | 0.616 |
| Primigest *vs.* multigest women | | 1,480 | [0.97; 2.257] | 0.069 | 1.839 | [0.932; 3.629] | 0.079 |
| Living site | Akodeha | | | | | | |
| | Ouedeme Pedah | 1.756 | [1.195; 2.582] | **0.004** | 0.788 | [0.429; 1.447] | 0.443 |
| | Comé | 0.430 | [0.308; 0.6] | **<0.001** | 0.404 | [0.222; 0.733] | **0.003** |
| Maternal education[f] | None | | | | | | |
| | Partial primary | 0.997 | [0.688; 1.444] | 0.987 | 1.373 | [0.752; 2.507] | 0.302 |
| | Complete primary | 1,285 | [0.741; 2.227] | 0.371 | 1.744 | [0.735; 4.138] | 0.207 |
| | Beyond primary | 0.651 | [0.404; 1.047] | 0.077 | 0.838 | [0.363; 1.932] | 0.678 |
| Number of visits (ANC + emergency)[g] | [0–4 | | | | | | |
| | [5,6] | 2.075 | [1.283; 3.355] | **0.003** | 4.679 | [1.498; 14.62] | **0.008** |
| | [7–12] | 3.023 | [1.86; 4.912] | **<0.001** | 6.127 | [1.958; 19.172] | **0.002** |
| Dry season | | 0.859 | [0.643; 1.148] | 0.305 | 1.120 | [0.691; 1.816] | 0.646 |
| Bednet possession | | 1.119 | [0.813; 1.539] | 0.491 | 0.943 | [0.547; 1.624] | 0.831 |
| Number of IPTp-SP doses[h] | [0,1] | | | | | | |
| | [2,3] | 0.642 | [0.379; 1.088] | 0.100 | 0.984 | [0.316; 3.066] | 0.977 |

[a]: malaria infection was defined by at least one positive TBS during pregnancy (follow-up or delivery); 365 women had at least one malaria infection during the follow-up.

[b]: an Odds Ratio <1 shows a negative association between the variable and malaria infection whereas an Odds Ratio >1 shows a positive association.

[c]: significant P value <0.05 is in bold.

[d]: Toxoplasmosis serological status defined at inclusion (n = 948).

[e]: age has been divided into 3 periods.

[f]: maternal education was sequenced into 4 categories.

[g]: number of visits has been divided into 3 categories.

[h]: number of IPTp-SP doses has been classified into 2 categories.

The logistic analysis was adjusted for maternal age, gravidae, living site, maternal education, number of visits, dry season, bednet possession and number of IPTp-SP doses.

Lastly, *T. gondii* seropositive women tended to have lower anti-*Pf*AMA1 IgG levels both at inclusion (*P* = 0.05) and at delivery (*P* = 0.002) compared to seronegative ones (Fig 4). In order to ensure the robustness of this association, a multivariate analysis adjusted for possible confounders was performed for anti-*Pf*AMA1 IgG levels at inclusion and at delivery (Table 4). As expected, malaria infections during the follow-up were associated with increased levels of anti-*Pf*AMA1 IgG in mothers at delivery. Also, and independently of malaria infection, anti-*Pf*AMA1 IgG levels were lower at delivery for *T. gondii* seropositive women compared to seronegative ones (*P* = 0.008; Table 4). A second analysis was performed to clarify when the levels of anti-*Pf*AMA1 IgG differ between *T. gondii* seropositive and seronegative pregnant women. Again, as expected, between inclusion and delivery, the levels of anti-*Pf*AMA1 IgG increased if women were infected by *P. falciparum* during the follow-up. Independently of malaria infections, the levels of anti-*Pf*AMA1 IgG decreased more sharply in *T. gondii* seropositive women than in seronegative women (S2 Table).

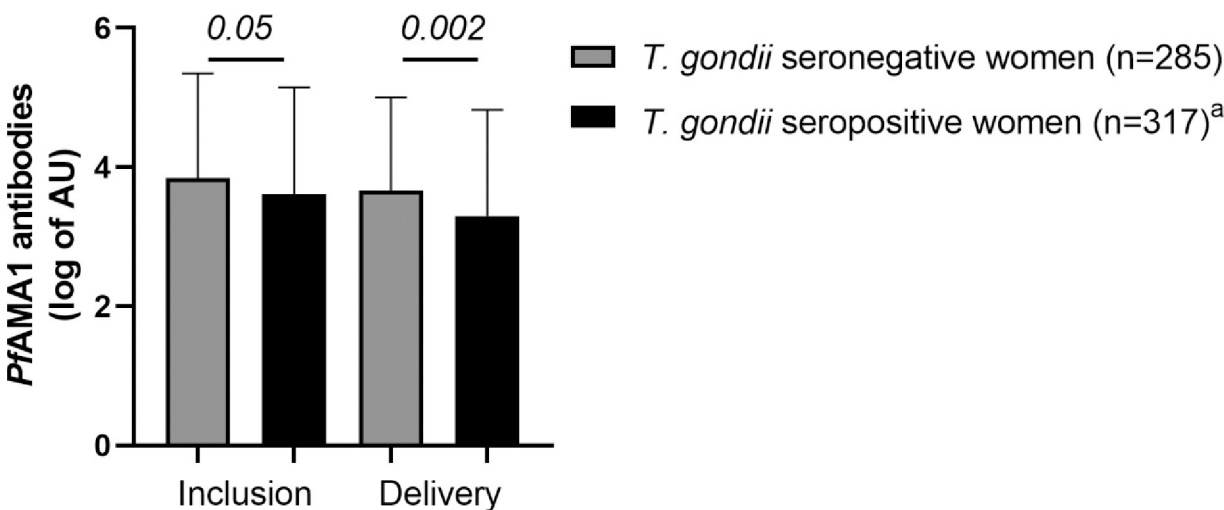

**Fig 4. Maternal IgG levels to *Pf*AMA1 in relation to *T. gondii* serological status.** Anti-*Pf*AMA1 IgG levels measured in STOPPAM study were compared according to maternal toxoplasmosis serological status. Only women with complete results (anti-*Pf*AMA1 and anti-*T. gondii* IgG at inclusion or delivery) were taken into account for the calculation. The bars represent the mean of anti-*Pf*AMA1 IgG levels, and the whiskers represent the standard deviations. [a]: 13 missing values at delivery.

## Discussion

To provide additional data on toxoplasmosis during pregnancy in Benin, we carried out a retrospective sero-epidemiological study on a longitudinal cohort of pregnant women. Performed between 2008 and 2010 on 974 women living in a rural area of southern Benin, this study showed a toxoplasmosis seroprevalence rate of 52.6% with a seroconversion rate of 3.4% among seronegative women, also interpretable as 1.4% among all women screened (value used subsequently in order to have the same comparison basis as that used in bibliographic references cited below), and a congenital toxoplasmosis rate of 0.2% among live births.

Although the follow-up of the cohort started early in pregnancy, with a mean gestational age of 17 weeks at inclusion [29], seroconversion and congenital toxoplasmosis rates were potentially underestimated for three main reasons. Firstly, the follow-up design began at the first ANC around the fourth month of pregnancy, so it cannot be excluded that seroconversion cases having occurred during the periconceptional period or at the beginning of pregnancy went unnoticed, especially since we did not measure the anti-*T. gondii* IgM. However, the risk of not having detected these cases is low since during this period, toxoplasmosis infections are rarely involved in maternal-fetal transmission and, if transmitted, frequently result in spontaneous miscarriage or malformations that would have been detected during ultrasound monitoring performed throughout the study [34]. Secondly, the depletion in some samples at delivery, as well as the lack of some samples due to miscarriages or stillbirths, contributed surely to an underestimation of the number of *T. gondii* seronegative women having seroconverted. Lastly, the non-monitoring of serological and ophthalmological follow-up of infants during their first year of life may have hampered the detection of tardive CT.

Despite these potential biases, the 1.4% seroconversion rate observed in the study aligns with the 1.6% value found overall in sub-Saharan Africa [14]. On a smaller scale in Benin, this rate of 1.4% in 2008–2010 is significantly higher than the values resulting from successive studies carried out later on smaller numbers of women, namely in 2011 (30% seroprevalence and 0.4% seroconversion rates among 283 women screened in the northern rural area) [25], in 2012 (48% seroprevalence and 0.7% seroconversion rates among 266 women screened in the

**Table 4. Multivariate analysis exploring the association between anti-PfAMA1 IgG levels at inclusion and delivery and *T. gondii* maternal serological status.**

| Independent variables | Categories | anti-*Pf*AMA1 IgG at inclusion (n = 650) [a] | | | anti-*Pf*AMA1 IgG at delivery (n = 627) [a] | | |
|---|---|---|---|---|---|---|---|
| | | Coef[b] | [95% CI] | *P* value[c] | Coef[b] | [95% CI] | *P* value[c] |
| *T. gondii* positive serological status | | -0.141 | [-0.372; 0.09] | 0.230 | -0.295 | [-0.514; -0.075] | **0.008** |
| Malaria infection[d] during follow-up | | | | | 0.337 | [0.107; 0.566] | **0.004** |
| Adjustment variables: | | | | | | | |
| Maternal age[e] | [15–22] | | | | | | |
| | [23–29] | -0.081 | [-0.395; 0.232] | 0.610 | -0.023 | [-0.318; 0.273] | 0.881 |
| | [30–35] | -0.499 | [-0.827; -0.17] | **0.003** | -0.230 | [-0.543; 0.084] | 0.151 |
| Primigest *vs.* multigest women | | -0.092 | [-0.446; 0.262] | 0.609 | -0.030 | [-0.354; 0.295] | 0.858 |
| Living site | Akodeha | | | | | | |
| | Ouedeme Pedah | 0.646 | [0.346; 0.947] | **<0.001** | -0.371 | [-0.659; -0.083] | **0.012** |
| | Comé | -0.479 | [-0.742; -0.217] | **<0.001** | -1.156 | [-1.412; -0.9] | **<0.001** |
| Maternal education[f] | None | | | | | | |
| | Partial primary | -0.258 | [-0.56; 0.043] | 0.093 | -0.222 | [-0.509; 0.064] | 0.128 |
| | Complete primary | | [-0.326; 0.584] | 0.577 | 0.168 | [-0.258; 0.595] | 0.438 |
| | Beyond primary | 0.129–0.300 | [-0.667; 0.067] | 0.109 | -0.149 | [-0.489; 0.191] | 0.389 |
| Number of visits (ANC + emergency)[g] | [0–4] | | | | | | |
| | [5,6] | | | | -0.096 | [-0.518; 0.326] | 0.656 |
| | [7–12] | | | | -0.104 | [-0.532; 0.325] | 0.635 |
| Dry season | | 0.117 | [-0.112; 0.347] | 0.316 | 0.019 | [-0.199; 0.238] | 0.863 |
| Bednet possession | | 0.025 | [-0.228; 0.279] | 0.844 | -0.028 | [-0.268; 0.212] | 0.821 |
| Number of IPTp-SP doses[h] | [0,1] | | | | | | |
| | [2,3] | | | | 0.008 | [-0.509; 0.524] | 0.977 |

[a]: Data on age and/or toxoplasmosis status and/or *Pf*AMA1 dosage were missing for 387 women at inclusion and for 410 women at delivery.

[b]: a coefficient <0 shows a negative association between the variable and malaria infection whereas a coefficient >0 shows a positive association.

[c]: significant P value <0.05 is in bold.

[d]: malaria infection was defined by at least one positive TBS during pregnancy (follow-up or delivery).

[e]: age has been divided into 3 periods.

[f]: maternal education was sequenced into 4 categories.

[g]: number of visits has been divided into 3 categories.

[h]: number of IPTp-SP doses has been classified into 2 categories.

The linear analysis was adjusted for quantitative TBS, maternal age, gravidae, living site, maternal education, number of visits, dry season, bednet possession and number of IPTp-SP doses.

southern urban area) [23] and in 2016 (36% seroprevalence and 0.5% seroconversion rates among 399 women screened in the southern rural area) [26]. Some factors can explain these differences: 1) two of the cited studies focused on toxoplasmosis, and therefore seronegative women were made aware of the risk of infection through the adoption of preventive measures [23, 26]; and 2) spatial and temporal heterogeneity in the distribution of the parasite may have occurred, as described in the introduction for Benin, and in agreement with observations reported in the annual reports of the French National Reference Center for Toxoplasmosis [35]. In view of these factors, and knowing that different serological techniques were used, the higher seroconversion rate found in our study can also be explained by its retrospective aspect: the original women's follow-up focused on malaria without any attention being paid to toxoplasmosis, so this infection followed its natural course without any preventive measures. The absence (except in one case) of specific anti-*T. gondii* IgM has not been explained. It may be due to their lability, i.e., transient or short time presence, during the specific IgG positivation

period [36]. However, an absence of specific IgM to toxoplasmosis is not uncommon, with no precise explanation provided [37].

In terms of public health, it is] ing to consider the situation with respect to other countries that manage toxoplasmosis in pregnancy differently. During the same 2008–2010 period as considered here, the United States, where no routine toxoplasmosis gestational screening was conducted, reported a low toxoplasmosis seroprevalence rate of 9% [38] associated with high CT rates ranging from 0.01% to 0.1% depending on the states [39, 40]. In comparison, France reported a higher 37% seroprevalence rate associated with a lower 0.02% CT rate, as an outcome of its toxoplasmosis gestational screening [41]. In Benin, where the screening is limited to hospitals or large private clinics and is costly [23, 26], one would expect the high prevalence rate of 52.6% to be associated with a much higher rate of CT than in the United States where the circulation of the parasite is low. But the fairly close value of 0.2% is compelling and suggests a possible reducing effect on *in utero T. gondii* transmission, which could be attributable to the IPTp-SP administered during pregnancy, a treatment that is inexistent in non-tropical countries such as in the United States.

Investigating the impact of IPTp-SP on *T. gondii* infection is not new [37]. Despite ethical impossibilities precluding experiments on it, we can hypothesize that IPTp-SP administered against malaria could reduce the impact of pregnancy-associated toxoplasmosis on fetuses. The doses would not prevent infection but would decrease the serious parasitic effects. Currently, IPTp-SP is administered as often as possible from the second trimester of pregnancy in malaria endemic areas [27], and WHO currently recommends at least three SP doses during pregnancy. When the STOPPAM project was implemented in 2008–2010, the current national guidelines were followed and, under the supervision of midwives, 86% of the women received two SP doses (each dose corresponding to 1,500mg sulfadoxine and 75mg pyrimethamine) spaced at least one month apart from the second trimester of pregnancy [42–44]. In parallel, fetal ultrasonography was performed three times at gestational weeks 26, 30 and 36 [32] and did not reveal any abnormality of the cephalic structure among the fetuses from the seroconversion maternal group. Similarly, no birth defects were observed. Despite this low number of observations, it is tempting to hypothesize that the IPTp-SP may have contributed to limiting the severe adverse consequences due to toxoplasmosis, as in the case of PW2. To illustrate this hypothesis, even if the aim of the study was different, a multicenter randomized trial was conducted to compare the efficacy and tolerance of pyrimethamine (50mg/day) associated with sulfadiazine (3x1g/day) and folinic acid versus spiramycin (3x1g/day), in order to reduce *T. gondii* placental transmission [45]. Although few individuals were in the compared groups due to the interruption of the trial, this study found a trend toward lower *T. gondii* placental transmission in the SP group, in which no fetal cerebral toxoplasmosis lesions were observed. Thus, although the sulfamide-pyrimethamine regimens were very different between this latter study and ours (since the parasite targets were different), it led to a reduced severity of the fetal sequelae, which is in agreement with the absence of ultrasonographic signs in our PW2 case despite a maternal infection having occurred at around 4 months of gestation.

Strikingly, this study observed a negative relationship between *T. gondii* seropositivity and both the number of malaria infections during pregnancy and the presence of placental malaria at delivery. Paradoxically, this finding was accompanied by lower anti-*Pf*AMA1 IgG levels in *T. gondii* seropositive women, compared to others, especially at delivery. Would *T. gondii* presence provide protection to the host against malaria? Several elements may be proposed to explain this:

Seropositivity to *T. gondii* reflects not only a past infection with this parasite but also the maintenance of a control immunity against the encysted forms of the parasite, mainly in the brain and muscle tissues [46]. On the other hand, the AMA1 protein is a conserved element of

*Apicomplexa* parasites [47–49] (with about thirty percent homology between *T. gondii* and *Plasmodium*) for which it would play an essential role in the parasite positioning on the host cell's membrane for favoring cell invasion [47]. Thus, toxoplasmosis infection could generate anti-AMA1 IgG potentially cross-reacting with *Pf*AMA1 antigen sites. Consequently, this could either down-regulate the production of specific anti-*Pf*AMA1 IgG necessary to fight against malaria or slow down/prevent the formation of the tight attachment zone between *Plasmodium* and the host cell, created by a parasitic protein complex [47]. Indeed, the production of anti-AMA1 Ig would block the interaction between RON (rhoptry neck protein) and AMA1 localized in the moving junction necessary for parasite invasion [50, 51]. To further investigate this hypothesis, it would be necessary to measure anti-*Tg*AMA1 IgG levels and to introduce these new data in the global analysis.

Another explanation could be that the persistent installation of *T. gondii* in its host involves a dynamic regulation of combined cellular and molecular immunoregulatory networks [52–55], which can impact the newly infecting *Plasmodium* parasite using similar pathways. This could result in less production of anti-*Pf*AMA1 IgG to effectively fight plasmodial parasites. This hypothesis seems paradoxical due to the clearly demonstrated association between anti-*Pf*AMA1 IgG levels and the clinical and parasitological protection against malaria [56]. Nevertheless, it may be consistent in that these IgG levels, measured at delivery, reflected a lower need for *T. gondii* seropositive women to react against *P. falciparum* infection during their pregnancy.

Finally, and without excluding the previous explanations, another suggestion is based on experimental observations where mice primo-infected with *P. berghei*, treated with antimalarial drugs and then infected with *T. gondii*, developed immune benefits such as faster production of anti-*T. gondii* IgG in comparison to mice infected with *T. gondii* alone [53].

Altogether, these possible explanations may support the observation in this study of better immune control towards malaria for pregnant women chronically infected by *T. gondii*.

Except for three previous studies—two of which used PCR methods on Ghanaian pregnant women only at delivery [28] or at the recruitment on the third trimester of pregnancy [37] and the third study investigating hematological parameters in young children in Cameroon [57]—to our knowledge, this study is the first to simultaneously investigate *T. gondii* and *P. falciparum* in a longitudinal human cohort. This retrospective sero-epidemiological study on toxoplasmosis in Benin was primarily dedicated to drawing the medical staff's attention to this pathology, which is underestimated during pregnancy in African countries where other infectious pressures such as that exerted by malaria is extremely concerning. For the first time in Benin, it offered the possibility of investigating a cohort of nearly 1,000 pregnant women. Although this observation goes back 10 years, the high CT rate concluded in the findings reinforces the importance of implementing preventive measures as much as possible against toxoplasmosis to pregnant women in the absence of knowledge of their serological status and when no screening program exists. Indeed, this study also put forward that women with a low education level were more likely to be seropositive to *T. gondii*. This aspect, also highlighted in a study in the United States [58], is undoubtedly the cause and the consequence of less access to and less awareness of prevention messages among these women.

This study also concluded that a preexisting infection by *T. gondii* plays a potential protective role against malaria. It follows that it is essential to understand the interactions between pathogens infecting a single human host, in order to properly measure their effects, which are neither strictly protective nor strictly deleterious but imbalanced. In this sense, further investigations on the cross-reactivity between IgG specific for *T. gondii* and *P. falciparum* would be necessary in groups representative of the general population in malaria endemic areas, in order to remove any bias linked to the IPTp delivered to pregnant women.

## Supporting information

**S1 Table. Multivariate analysis of effect of toxoplasmosis serological status and IPTp on malaria infection at delivery.**
(DOCX)

**S2 Table. Multivariate analysis of differences in anti-*Pf*AMA1 IgG levels between inclusion and delivery.**
(DOCX)

## Acknowledgments

We thank the teams of the microbiology and virology units from the Saint Louis Hospital as well as the virology team from the Bichat-Claude Bernard Hospital in Paris for welcoming us. We are particularly grateful to the team of the parasitology unit from the Bichat-Claude Bernard Hospital for their warm welcome and assistance. This study was part of the STOPPAM "Strategy To Prevent Pregnancy-Associated Malaria" collaborative project. We are grateful to all the women who participated in the study. Finally, we thank Sharon Calandra for her proofreading.

## Author Contributions

**Conceptualization:** Magalie Dambrun, François Simon, Sandrine Houzé, Florence Migot-Nabias.

**Data curation:** Magalie Dambrun, Célia Dechavanne, Valérie Briand.

**Formal analysis:** Magalie Dambrun, Célia Dechavanne, Valérie Briand, Florence Migot-Nabias.

**Funding acquisition:** Florence Migot-Nabias.

**Investigation:** Magalie Dambrun, Nicolas Guigue, Tristan Candau, Murielle Lohezic, Saraniya Manoharan, Nawal Sare, Firmine Viwami.

**Methodology:** Magalie Dambrun, Nicolas Guigue, François Simon, Sandrine Houzé, Florence Migot-Nabias.

**Project administration:** Magalie Dambrun, Florence Migot-Nabias.

**Resources:** Magalie Dambrun, Nadine Fievet, Florence Migot-Nabias.

**Supervision:** Florence Migot-Nabias.

**Validation:** Magalie Dambrun, Célia Dechavanne, Nicolas Guigue, Valérie Briand, Sandrine Houzé, Florence Migot-Nabias.

**Writing – original draft:** Magalie Dambrun, Célia Dechavanne, Sandrine Houzé, Florence Migot-Nabias.

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
