## [Decision Letter · Decision Letter 0]

22 Jun 2021

PONE-D-21-15174

Retrospective sero-epidemiological study of toxoplasmosis during pregnancy in Benin and potential relationships with malaria

PLOS ONE

Dear Dr. Dambrun,

Thank you for submitting your manuscript to PLOS ONE. After careful consideration, we feel that it has merit but does not fully meet PLOS ONE’s publication criteria as it currently stands. Therefore, we invite you to submit a revised version of the manuscript that addresses the points raised during the review process.

Two expert external reviewers liked your manuscript, but recommended that it be revised prior to acceptance. You'll see that reviewer #1 had many comments, while reviewer #2 had less, so due to the number of comments, rather the difficulty in replying to them, I have marked your submission down for major, rather than minor, revision.

We look forward to receiving your revised manuscript.

Kind regards,

Gordon Langsley

Academic Editor

PLOS ONE

“ We thank the Institut de Médecine et d’Epidémiologie Appliquée (IMEA) for financial support (grant 0602DIRmba).”

“FMN : grant 0602DIRmba; Institut de Médecine et d’Epidémiologie Appliquée (IMEA); https://www.imea.fr; no role of the funder in this study

NF : contract number 200889; European Union Framework 7; https://cordis.europa.eu/project/id/200889/fr; the funder had a role of data collection in this study”

Additional Editor Comments (if provided):

Reviewers' comments:

Reviewer's Responses to Questions

**Comments to the Author**

1. Is the manuscript technically sound, and do the data support the conclusions?

Reviewer #1: Yes

Reviewer #2: Partly

2. Has the statistical analysis been performed appropriately and rigorously? 

Reviewer #1: Yes

Reviewer #2: I Don't Know

3. Have the authors made all data underlying the findings in their manuscript fully available?

Reviewer #1: No

Reviewer #2: Yes

4. Is the manuscript presented in an intelligible fashion and written in standard English?

Reviewer #1: Yes

Reviewer #2: Yes

5. Review Comments to the Author

Reviewer #1: This is an interesting paper providing insights into the prevalence of toxoplasmosis in pregnant women and the incidence of congenital toxoplasmosis in Benin. The study is well-conducted and the paper reads very well. However, I have some important comments and some questions remain to be addressed.

First, the title is not self-explanatory and does not fully reflect the content of the paper. Maybe try: “Retrospective study of prevalence of toxoplasmosis in pregnant women in Benin and its relation with malaria diagnosis at delivery” ?

Results:

Line 161: the occurrence of a “positive (result) for the first assay and negative for the second”, is not straightforward, as it is mentioned (line 151) that only seronegative women were re-tested ate delivery. Please explain how this sequence of results could happen.

Line 206: what do you mean by “depleted”? Were these 49 samples negative when tested diluted to 1/3?

Line 267: I don’t share this viewpoint. The rate of seroconversion should be calculated on the seronegative only. Seropositive at first serology cannot “seroconvert”.

Line 271: it would be interesting here to convert this ratio into x/1000 live births, which is a widely rate used to compare incidence of diseases.

Line 278: it is not adapted to cite the gestational age here, especially as seropositive women were enrolled at an earlier stage (and the means are very close, 16.1 and 16.8 !). If seropositive had been enrolled at a later stage, it could have been an issue, as they might have acquired infection in early pregnancy (which would have been unnoticed as IgM were not tested).

Line 293: were the prevalence of prematurity (1/12) and low birth weight (2/12) similar to that observed in the whole group of women?

Line 302: it is very surprising that IgM were detected in only 1/12 patients. As it is the hallmark of Toxoplasma acute infection, it casts doubt on the results of IgG. It cannot be excluded that the technique used for the first sample (Bioplex) lacks sensitivity. Were these samples re-analyzed using the WB Toxo IgGII to confirm that they were indeed negative at the beginning of pregnancy? This could also explain why ultrasound surveillance did not detect any abnormalities. By contrast to the authors’ hypothesis (lines 367-70), it is unlikely that long-term storage of sera led to an alteration of antibody, as banks of sera are widely used to evaluate new serological assays with no such issue. Besides, infection with no IgM is a rare event (see Fricker-Hidalgo, JCM 2013). Or maybe the authors wanted to cite this paper but linked another one (ref #37)? PW is probably not a seroconversion, as there is no move in antibody dosages, and should be probably excluded.

The authors report one infected neonate, following a maternal infection at around 4 months of pregnancy, it is all the more strange that no ultrasonographic signs were detected, as congenital infection at that stage is symptomatic. This should be discussed.

Table 4: were PfAMA1 antibody dosages at inclusion and at delivery paired (same patient)? If not it seems difficult to interpret the decrease.

How was congenital infection diagnosed? Was IgM assessed in the cord blood? Only 2 infants had WB profiles different from their mother, so it is important to know how diagnosis was made or excluded.

Fig. 3&4: use preferably the same color legend. In Fig.4, bars should be grouped in each category.

Discussion:

Line 383: “Investigating the impact of IPTp-SP on T. gondii infection”: this sentence should appear in the objectives, at the beginning of the manuscript.

Lines 397-401: should be deleted, as this trial cannot support, nor be compared to two intakes of sulfadoxine-pyrimethamine. Indeed, it was a clinical trial comparing sulfadiazine-pyrimethamine versus spiramycine, thus did not address the issue of treating or nor treating. Furthermore, threatment was given daily.

Line 432: “new antiparasitic compounds with anti-T. gondii and anti-P. falciparum cross-species efficacy could soon be defined”: such what? Antiparasitic drugs already target the same pathways. Maybe this sentence could be rephrased, if the authors rather think of immune cross-reactivity (vaccine development or immunomodulation therapy?).

How do the authors explain that Toxoplasma seropositivity had a preventive effect on malaria infection at delivery, but no overall effect during follow-up (table 3)?

Minor points:

Line 62 : prefer “inversely correlated to the age of pregnancy at maternal infection”

Line 72: sulfadoxine is no more available for the treatment of toxoplasmosis.

Reviewer #2: General comments

The manuscript presented Toxoplasma seroepidemiology and malaria infection in a large cohort of pregnant women in a region of Benin. The study is interesting and well-written. It showed the epidemiology of congenital toxoplasmosis in a region where is applied an intermittent preventive treatment against malaria during pregnancy using a sulfadoxine-pyrimethamine (SP) combination (IPTp-SP). However, based on the results of this observational study in a specific population, the authors tried to find a relation between toxoplasmosis and malaria.

Specific comments

Title

The data presented in the study did not show any relation between toxoplasmosis and malaria during pregnancy. In consequence, could the authors modify the title by suppressing « potential relationship »?

Abstract

Lines 37-39: Could the authors specify IgG are anti- P. falciparum Apical Membrane Antigen 1?

Mat & Meth

Lines 163-164: Could the authors reconsider this sentence?

According to the reviewer, this assay has been developed and been used to detect low titers of specific anti-Toxoplasma IgG. It is not a screening test, as some assays using a limited number of parasite antigens.

Results

Toxoplasmosis seroprevalence, seroconversion and CT

Could the authors give here or in Seroconversion maternal group and infants the % of CT among infants whose mothers had an infection during pregnancy?

Association between T. gondii serological status and P. falciparum infection during pregnancy

Lines 320-323: The OR is low 0.531 (< 2 fold fewer risks for Toxoplasma seropositive pregnant women to have malaria at delivery). Could the authors specify this in the section Discussion?

Table 3: Could the authors shortly present in the text the results of malaria riks according to the number of visits?

Discussion

Lines 350-351: The authors should add that the lack of opthalmological examination in infants in their study could lead to misdiagnosing CT cases.

Lines 410-412: Could the authors give the % of homology between the proteins of T. gondii and P. falciparum.

The authors should consider that the tendency to observe a lower malaria infection at delivery occurred in a population of women treated with sulfadoxine-pyrimethamine (SP) combination. To support a relation between toxoplasmosis and malaria such a study should be done in non-treated pregnant women and, or the general population. Indeed, we could not exclude that such treatment affected the risk of malaria between Toxoplasma negative and seropositive pregnant women. To support this last hypothesis, the rate of malaria was not different between Toxoplasma negative and seropositive women at the beginning of pregnancy (Figure 3).

Finally, the data are old (2008-2010, more than ten years).

6. PLOS authors have the option to publish the peer review history of their article (what does this mean?). If published, this will include your full peer review and any attached files.

Reviewer #1: No

Reviewer #2: No

---

## [Author Response · Author response to Decision Letter 0]

30 Jul 2021

Please find the responses to the Editor and to the Reviewers attached to the cover letter.

---

## [Decision Letter · Decision Letter 1]

31 Aug 2021

PONE-D-21-15174R1

Retrospective study of toxoplasmosis prevalence in pregnant women in Benin and its relation with malaria

PLOS ONE

Dear Magalie,

You improved your revision to the satisfaction of reviewer #2, but reviewer #1 still raised a number of minor points that appear easily addressed, so to give you the opportunity to address the points of reviewer #2 I have marked you ms dpown for "minor revision". If you make it absolutely clear in your rebuttal how you dealt with each point I should be able to make a rapid editorial decision without sending your further revised ms back out for review.,

Thank you for submitting your manuscript to PLOS ONE. After careful consideration, we feel that it has merit but does not fully meet PLOS ONE’s publication criteria as it currently stands. Therefore, we invite you to submit a revised version of the manuscript that addresses the points raised during the review process.

We look forward to receiving your revised manuscript.

Kind regards,

Gordon Langsley

Academic Editor

PLOS ONE

Journal Requirements:

Reviewers' comments:

Reviewer's Responses to Questions

**Comments to the Author**

1. If the authors have adequately addressed your comments raised in a previous round of review and you feel that this manuscript is now acceptable for publication, you may indicate that here to bypass the “Comments to the Author” section, enter your conflict of interest statement in the “Confidential to Editor” section, and submit your "Accept" recommendation.

Reviewer #1: (No Response)

Reviewer #2: All comments have been addressed

2. Is the manuscript technically sound, and do the data support the conclusions?

Reviewer #1: Yes

Reviewer #2: Yes

3. Has the statistical analysis been performed appropriately and rigorously? 

Reviewer #1: Yes

Reviewer #2: Yes

4. Have the authors made all data underlying the findings in their manuscript fully available?

Reviewer #1: No

Reviewer #2: Yes

5. Is the manuscript presented in an intelligible fashion and written in standard English?

Reviewer #1: Yes

Reviewer #2: Yes

6. Review Comments to the Author

Reviewer #1: The authors have addressed most comments, but some clarifications are still needed.

About M&M of serology:

- line 160: do you mean that when the Platelia IgG assay was positive at the end of pregnancy for a woman previously diagnosed as seronegative, both sera (inclusion and end) were retested with Platelia IgG? This is still not clear.

- line 163: for the “rare samples with confirmed discrepant results”, do you mean “negative with Bioplex and positive with Platelia at inclusion” ? or “samples at end of study that tested positive, then negative, with Platelia during the first and second run, respectively ” ?

- line 174: “qualitative” should be removed, as the Platelia assay is a quantitative assay.

Results:

- line 212: were any of these 49 women with a negative result obtained on diluted samples at inclusion, diagnosed seropositive at delivery? It should be mentioned, so that there is doubt regarding the 12 announced cases of seroconversion.

- line 269: precise n/N seronegative samples available at delivery

- line 274: precise “available plasma samples at delivery”

- line 275: if there were 12 mother-cord plasma samples, why were only 9 “available for highlighting potential cases of CT” ? It could be rephrased as “from the 12 cases identified with seroconversion during pregnancy, mother and cord blood paired samples were available in only 9”. For better understanding, please mention the results of Platelia Toxo IgM (0/9 positive samples ?) and of WB IgG/IgM, as described in M&M.

- line 277: regarding the rate of CT: 2/820 live births: does it mean that there has been 159 miscarriages? This should be discussed, as some fetal losses might have been due to unrecognized toxoplasmosis.

-the table presented in the response to reviewer 2 should be shown, at least as supplemental material.

Discussion:

-line 367: the rate of seroconversion is 3.4%, not 1.4%, as calculated in the Results section.

- line 383: I would like to insist that seroconversion with no IgM is a rare event, thus it not “not uncommon”, as written, and the sentence must be amended. The study by Fricker-Hidalgo reported 15 cases from 12 centers over 10 years, i.e. 15/4500 (0.3%) cases of seroconversion, which is far different from your results.

- lines 416-19: need to be rephrased. Maybe try “Thus, although the sulfamide-pyrimethamine regimens were very different between this latter study and ours, it led to a reduced severity of the fetal sequelae, which is in agreement with the absence of ultrasonographic signs in our PW2 case despite a maternal infection having occurred at around 4 months of gestation.”

Reviewer #2: The authors answered all the comments of the reviewer.

Minor comment

Lines 364-365: Ophthalmological follow-up (as serological follow-up) during the first year of life is sufficient to diagnose CT.

Long-term ophthalmological follow-up is useful to diagnose ocular toxoplasmosis in newborns previously diagnosed with CT.

7. PLOS authors have the option to publish the peer review history of their article (what does this mean?). If published, this will include your full peer review and any attached files.

Reviewer #1: No

Reviewer #2: No

---

## [Author Response · Author response to Decision Letter 1]

17 Sep 2021

The requested information is provided in the file including both the cover letter and the responses to reviewers.

---

## [Editor Report · Decision Letter 2]

18 Oct 2021

PONE-D-21-15174R2Retrospective study of toxoplasmosis prevalence in pregnant women in Benin and its relation with malariaPLOS ONE

Dear Dr. DAMBRUN,

Thank you for submitting your manuscript to PLOS ONE. After careful consideration, we feel that it has merit but does not fully meet PLOS ONE’s publication criteria as it currently stands. Therefore, we invite you to submit a revised version of the manuscript that addresses the points raised during the review process. Although referee 1 found your revised manuscript much improved a number of minor clarifications have been requested, so in your rebuttal please make absolutely clear how you addressed the points raised, as this will help me make a rapid editorial decision without sending your manuscript back out for review.

We look forward to receiving your revised manuscript.

Kind regards,

Gordon Langsley

Academic Editor

PLOS ONE
---

## [Author Response · Author response to Decision Letter 2]

2 Dec 2021

We are responding to the request addressed to us to send the latest version of our revised manuscript without changes (e-mail from the Editorial Manager on November, 24th). This last version corresponds to the R2 version sent on September, 17th. The only modification that the Editorial Manager will have to bring concerns the "Data Availability statement". To comply with the online submission procedure, we have entitled “Dambrun et al_Text” the clean version and “Dambrun et al_Text final” the version “with track changes”. These two texts are identical. We hope to respond as clearly as possible to the request of the Editorial Manager.

---

## [Editor Report · Decision Letter 3]

16 Dec 2021

Retrospective study of toxoplasmosis prevalence in pregnant women in Benin and its relation with malaria

PONE-D-21-15174R3

Dear Dr. MAGALIE DAMBRUN,

We’re pleased to inform you that your manuscript has been judged scientifically suitable for publication and will be formally accepted for publication once it meets all outstanding technical requirements.

Kind regards,

Gordon Langsley

Academic Editor

PLOS ONE
---

## [Editor Report · Acceptance letter]

30 Dec 2021

PONE-D-21-15174R3 

Retrospective study of toxoplasmosis prevalence in pregnant women in Benin and its relation with malaria 

Dear Dr. DAMBRUN:

I'm pleased to inform you that your manuscript has been deemed suitable for publication in PLOS ONE. Congratulations! Your manuscript is now with our production department. 

Kind regards, 

on behalf of

Dr. Gordon Langsley 

Academic Editor

PLOS ONE